# Changes in the Serum Metabolome in an Inflammatory Model of Osteoarthritis in Rats

**DOI:** 10.3390/ijms25063158

**Published:** 2024-03-09

**Authors:** Neus I. Berenguer, Vicente J. Sifre Canet, Carme Soler Canet, Sergi Segarra, Alejandra García de Carellán, C. Iván Serra Aguado

**Affiliations:** 1Departamento de Medicina y Cirugía Animal, Universidad Católica de Valencia San Vicente Mártir, 46002 Valencia, Spain; neusiris12@hotmail.com (N.I.B.); vj.sifre@ucv.es (V.J.S.C.); mdc.soler@ucv.es (C.S.C.); alejandra_vet@hotmail.com (A.G.d.C.); 2Centro de Investigación Translacional San Alberto Magno, Universidad Católica de Valencia San Vicent Mártir, 46002 Valencia, Spain; 3R&D Bioiberica SAU, Esplugues de Llobregat Barcelona, 08950 Barcelona, Spain; ssegarra@bioiberica.com

**Keywords:** osteoarthritis (OA), metabolomics, liquid chromatography/mass spectrometry (LC/MS), metabolic pathway, lipid molecules

## Abstract

Osteoarthritis (OA) is a pathology of great impact worldwide. Its physiopathology is not completely known, and it is usually diagnosed by imaging techniques performed at advanced stages of the disease. The aim of this study was to evaluate early serum metabolome changes and identify the main metabolites involved in an inflammatory OA animal model. This study was performed on thirty rats. OA was induced in all animals by intra-articular injection of monoiodoacetate into the knee joint. Blood samples were taken from all animals and analyzed by mass spectrometry before OA induction and 28, 56, and 84 days following induction. Histological evaluation confirmed OA in all samples. The results of this study allow the identification of several changes in 18 metabolites over time, including organic acids, benzenoids, heterocyclic compounds, and lipids after 28 days, organic acids after 56 days, and lipid classes after 84 days. We conclude that OA induces serological changes in the serum metabolome, which could serve as potential biomarkers. However, it was not possible to establish a relationship between the identified metabolites and the time at which the samples were taken. Therefore, these findings should be confirmed in future OA studies.

## 1. Introduction

Osteoarthritis (OA) is the most common form of arthritis and one of the most prevalent diseases in middle-aged and older people. Knee osteoarthritis is one of the leading causes of physical disability in adults [1,2,3,4]. Initially, it was seen as a disease in which only mechanical degradation of the cartilage occurred, but nowadays, it is considered a very complex disease involving different tissues [5,6]. Thus, alterations in the joint happen at different levels, namely, in the metabolism and architecture of the subchondral bone and in the morphology and metabolism of the articular cartilage, presenting periarticular osteophytosis, inflammation, meniscus degeneration, and fibrosis of the synovial membrane in different degrees. In addition, it is related to changes in other tissues, such as ligaments, tendons, and the sand surrounding the musculature [6,7,8,9,10]. Changes in the morphometric characteristics of the infrapatellar fat pad have also been observed, supporting an important role in the pathogenesis and progression of OA [6,11].

The epidemiology of this disorder is complex and multifactorial, with genetic, biological, and biomechanical components [6,7,9,10]. The main risk factors are age, obesity, sex, abnormal mechanical joint loading, altered joint morphology, and previous joint injury, especially previous knee injuries [7,11,12,13,14,15,16].

Despite its worldwide importance, there is no official treatment that cures, reverses, or slows down the development of OA. This could be because its pathogenesis is not yet fully understood [1,17]. For this reason, treatment of OA has traditionally consisted of management of the primary cause, followed by treatment of pain, control of clinical signs, and surgical intervention in some late stages [1,7,18].

For OA diagnosis, a new tool called metabolomics has emerged, and it might provide more valuable information, given that current imaging techniques offer a late diagnosis lacking information on the functional adaptation of cartilage. Metabolomics consists of the study of small biological molecules in a system and holds great potential for early diagnosis, monitoring therapies, and the understanding of the pathogenesis of many diseases [1,3,4,7,19].

For this reason, it has become an ideal method for the identification of OA biomarkers in a variety of biological samples. Different studies have reported several metabolites and metabolic pathways that can be altered in OA, such as amino acid metabolism, fatty acid and lipid metabolism, phospholipids, arginine, phosphatidylcholine, L-tryptophan, tyrosine, carnitine, and arachidonic acid [1,2,4,10,20,21,22,23,24,25]. In order to identify biomarkers, one must go to the metabolic pathways that affect amino acid metabolism. These include the biomarkers branched-chain amino acids (BCAAs), arginine, and phospholipid metabolism related to the conversion of phosphatidylcholine (PC) to lysophosphatidylcholine (lysoPC) [3]. In the study by Zhang et al., they associated six metabolites with knee OA: arginine, sphingomyelin, and different PC [4].

This current study has been performed through non-targeted metabolome and gene expression detection in samples obtained from rats using a monoiodoacetate (MIA)-induced OA model. The main objective was to evaluate the serum metabolome and identify the main altered metabolites in a patient with osteoarthritis of inflammatory origin. The hypothesis of the present study was that metabolomics could allow for the detection of changes in the serum metabolome in a patient with early osteoarthritis that have not yet been described.

## 2. Results

### 2.1. Metabolomic Study

Blood samples were taken from all animals and analyzed by mass spectrometry before OA induction (T0), 28 days (T28), 56 days (T56), and 84 days (T84) after OA induction.

A paired analysis of serum metabolite results at T0, T28, T56, and T84 was performed.

#### 2.1.1. T0 vs. T28 Paired Analysis

With the significant variables selected in the Volcano plot explained in material and methods, an Orthogonal Projections to Latent Structures Discriminant Analysis (OPLS-DA) multivariate analysis was carried out to detect the variables with the highest discriminant power.

Figure 1 shows how the model discriminates between the two times T0 and T28 in the score plot obtained after this analysis.

The model diagnostics were adequate with R2Y = 0.994 and Q2Y = 0.967, and the model is further validated with a *p*-value coefficient of variation (CV)-ANOVA < 0.001 and a permutation test of 1000 iterations (Figure 2).

The variable importance in the projection plot (VIP Plot) was performed from the model to select the most important discriminant variables.

From the VIP Plot, those variables with a low interaction coefficient (CI) (not including 0), which were among the first 30 variables ranked by discriminant order (VIP score), were selected. Extracting each ion (*m*/*z*) from one of the quality control (QC) raw data and checking the peak shape and retention time also verified each variable.

Table 1 shows the selected variables for identification using the human metabolome database (HMDB) [26,27] and Metlin databases [28], as well as the mass spectrometry (MS/MS) analyzed on the equipment.

Eight organic acids and derivatives, benzenoids, organoheterocyclic compounds, and lipid molecule variables were selected. These include lactacystin (C_15_H_24_N_2_O_7_S) from the carboxylic acid class and derivatives; Taurine (C_2_H_7_NO_3_S) from the organic sulfonic acids class and derivatives; Styrene oxide (C_8_H_8_O) and Tyramine (C_8_H_11_NO) from the benzene class and substituited derivatives; Setanaxib (C_21_H_19_ClN_4_O_2_) from the pyridines class and derivatives; Norsalsolinol (C_9_H_11_NO_2_) from the tetrahydroisoquinolines class; Ganoderic acid V (C_32_H_48_O_6_) from the prenol lipids class; and Ganglioside GM1 (C_72_H_130_N_2_O_31_) from the class sphingolipids (Table 2).

#### 2.1.2. T0 vs. T56 Paired Analysis

In this case, the significant variables were only five variables, which were selected to build the OPLS-DA model, whose score plot is presented in Figure 3.

As can be seen in the graph, samples 1, 4, 8, and 11 seem to form a cluster between them, which indicates that there are other variables not considered, apart from time, that may be influencing the results (intra-group variability).

Despite this, the VIP plot is constructed to order the variables by discriminant power and select the most important ones to identify them. Out of six variables, one of them was discarded because the confidence interval of the VIP score was too high, and another one because the chromatographic peak was not verified with the retention time.

Table 3 shows an abridgment of the results obtained for the four variables finally obtained.

The identification results of the selected variables were three organic acids and derivatives, including gamma-Glutamylcysteine (C_8_H_14_N_2_O_5_S), tyrosyl-Serine (C_12_H_16_N_2_O_5_), and acetyl citrate (C_8_H_10_O_8_) from the carboxylic acid class and derivatives (Table 4).

#### 2.1.3. T0 vs. T84 Paired Analysis

With the significant variables selected in the previous analysis, to detect the variables with the greatest discriminating power, an OPLS-DA multivariate analysis was carried out.

Figure 4 shows how the model discriminates between the two times T0 and T84 in the score plot obtained after this analysis. The sample of animal 8 (T84) must be considered as it differs from the rest of the samples in that group.

The model diagnostics were adequate with R2Y = 0.946 and Q2Y = 0.925, and the model is further validated with a *p*-value CV-ANOVA < 0.001 and a permutation test of 1000 iterations (Figure 5).

From the VIP Plot, those variables with a low CI (not including 0) and which were among the first 30 variables ranked by discriminant order (VIP score) were selected. Extracting each ion (*m*/*z*) in one of the QC raw data and checking the peak shape and retention time also verified each variable.

Table 5 presents the variables finally selected for identification using the human metabolome database (HMDB) [26,27] and Metlin databases [28], as well as the mass spectrometry (MS/MS) analyzed on the equipment.

The identification results of the selected variables were seven lipid molecules, including ginsenoside Rh1 (C_36_H_62_O_9_) and theasapogenol A (C_30_H_50_O_6_) from the prenol lipids class; phosphatidic acid (C_37_H_71_O_8_P) from the glycerophospholipids class; polyporusterone F (C_28_H_46_O_5_), brassinolides (C_27_H_46_O_6_), ursodeoxycholic acid (C_24_H_40_O_4_) from the steroids class and steroid derivatives; and 10-hydroperoxy-H4-neuroprostane (C_22_H_32_O_6_) related to prostaglandins and from the fatty acyls class (Table 6).

From the results obtained, it can be concluded that there are differences between the times T0 and T84.

In the discriminant analysis, we observed intra-group variability, which indicates that there are other variables not contemplated in the study, in addition to time, that are influencing the distribution of metabolites.

### 2.2. Histologic Study

As expected, all rats developed degenerative and inflammatory changes associated with OA after the MIA injection. This occurred at every time point after 28, 56, and 84 days (Figure 6 and Figure 7).

Histologic signs of osteoarthritis were observed microscopically in all OA-induced samples. The main alterations presented were the reduction in the cartilage stain intensity and the presence of an irregular cell density along the samples (Table 7) (Figure 8 and Figure 9).

## 3. Discussion

OA induced in rats by intra-articular MIA produces changes in the serum metabolome, as observed in the results of the study described herein. Furthermore, these variations are associated with the time elapsed since the induction of OA.

The pathophysiology of the OA complex remains unknown nowadays, so it was decided to investigate metabolomics as an advanced diagnostic method to see whether it could provide further insights into the disease. This technique allows us to observe the progress at a biochemical level and monitor the evolution of the treatment [4,10,17,19]. In addition, as the study period was relatively short, we needed a sampling technique that would allow us to make an early diagnosis compared to other techniques, such as radiography or MRI, where the diagnosis is made in more advanced stages of the disease [1,7,10,29,30]. The usefulness of an adequate OA model was verified by histological examinations.

Metabolomics is supported by several studies in which significant differences in serum metabolites were observed between healthy rats and OA rats, as observed in this study [1,2,10,17,19,20]. In the study by Chen et al. in rats with OA, plasma samples were analyzed by metabolomics, and it was shown that density labeling mass spectrometry, the same technique used in this study, had a high sensitivity for detecting metabolites in rat plasma [1].

On the other hand, one of the factors to consider was the use of serum to analyze the metabolites present in the animal. The study carried out by Zhang et al. compared blood and synovial fluid (SF), observing how the range of metabolites varied considerably, and out of 168, only 8 were consistently related. This study suggests that metabolic changes are joint-specific and other inflammatory processes may influence the concentration of metabolites in serum [4], as does the study carried out by Guma et al., which states that there is a fairly modest correlation between plasma and SF [19].

Thus, as discussed in several articles, it is necessary to know which metabolites are altered for a better understanding of the joint status [3,19,31,32]. Regarding the metabolites found in this study, they differ depending on whether the sample has been collected at T28, T56, or T84. Different benzenoids, organoheterocyclic compounds, organic acids, and lipid molecules were detected at T28, whereas only organic acids were observed at T56, and at T84, they were mainly lipid molecules.

The differential benzenoid classes found in this study were benzene and substituted derivatives. The organoheterocyclic compound classes found in this study were tetrahydroisoquinolines, pyridines, and derivatives. Previous studies had not observed any association of these metabolites with OA.

Likewise, differences in organic acids and derivatives were detected at T0 vs. T28 and T0 vs. T56. The main organic acid classes found in this study were carboxylic acids and derivatives (T28 and T56) and organic sulfonic acids and derivatives (T28). The organic sulfonic acid detected in the analysis at T0 vs. T28 was taurine. This result correlates with other studies where taurine metabolism was found to be one of the metabolic pathways most involved in OA, as taurine is implicated in the pathophysiology of OA, correlating with subchondral bone sclerosis and playing a vital regulatory role [21,29,30,33,34,35]. Anderson et al. observed that elevated taurine in OA could indicate increased subchondral bone sclerosis [35]. Yang et al. showed that taurine levels in sclerotic subchondral bone were positively regulated [30]. Taken together, these studies in synovial fluid revealed that altered taurine metabolism in subchondral bone has a direct correlation with subchondral bone sclerosis in osteoarthritis. In this study, taurine was analyzed from blood samples, so this metabolite could be used as an early biomarker of subchondral bone sclerosis in OA, but further blood studies are required to confirm this hypothesis.

Within the organic acid group, we have also detected carboxylic acids at T28 and T56, just as Swank et al. detected hippurate carboxylic acid in urine after 18 months of OA progression [21]. Within this group, we found that at T56, the metabolite acetyl citrate was detected. With regard to this metabolite, different studies have detected the presence of citrate in urine and synovial fluid related to OA [35,36,37]. Citrate is an intermediate in the tricarboxylic acid cycle, and its increase indicates an OA-related alteration in the cycle [33,35,36,37,38]. Therefore, it would be interesting to further investigate the presence of acetyl citrate in the blood as a biomarker of OA.

Lipid molecules were found at T0 vs. T28 and T0 vs. T84. These possible biomarkers were related to the findings of several studies, where there is an alteration of lipid metabolism associated with OA due to its pro-inflammatory properties [22,23,24,39,40]. Kosinska et al. detected alterations at the level of phospholipids and sphingolipids at different stages of the disease in synovial fluid [41,42,43]. Thus, understanding the relationship between OA and lipid molecule analysis may be helpful in future treatments [5].

The main lipid classes in which differences were found in this study are sphingolipids at T28, prenol lipids at T28 and T84, and glycerophospholipids, steroids, and fatty acyls at T84. These data are consistent with the study by Pousinis et al. that describes the presence of glycerophospholipids, sphingolipids, and fatty acyls in plasma from rats subjected to an OA model for 112 days [40].

With regard to prenol lipids, their most biologically relevant classes are fat-soluble vitamins (vitamins E, A, and K) [44]. Neogi et al. observed an association between low plasma levels of vitamin K and an increased prevalence of OA manifestations in the hand and knee. They found the relationship because vitamin K supports calcium homeostasis and facilitates bone mineralization [45,46,47]. Regarding vitamin E, different studies have demonstrated its potent anti-inflammatory properties as well as its role in the prevention and regulation of the progression of age-related diseases [48,49]. Therefore, further research on the relationship between OA and prenol lipids would be desirable.

Sphingolipids detected at T28 had been previously detected in other studies and were related to subchondral bone sclerosis in OA. This fact suggests that sphingolipids play an important regulatory role in the pathological process of sclerotic subchondral bone [24,30,40,42]. Tootsi et al. found changes in serum sphingolipid levels in humans with OA, confirming their involvement in the pathogenesis of OA [24]. Kosinska et al. found that sphingolipids could alter synovial inflammation and repair responses in damaged joints [42]. Thus, it would be interesting to use sphingolipids as blood biomarkers for OA.

Phospholipids are molecules associated with inflammation and increased cartilage damage at the synovial fluid level and may be associated with the pathogenesis of OA [41]. The glycerophospholipids form the essential lipid bilayer of all biological membranes, and changes in glycerophospholipid concentrations and composition are associated with OA development, as shown in a multitude of studies [4,22,24,40,50]. Therefore, changes in the concentration of lipid molecules, more specifically glycerophospholipids, may indicate a risk of OA.

The biomarker ursodeoxycholic acid detected in this study belongs to the class of steroids, superclass lipids, and lipid-like molecules (HMDB). Ursodeoxycholic acid is a naturally occurring dihydroxy hydrophilic bile acid, and Moon et al. demonstrated that this bile acid has a preventive potential as a treatment in a model of induced OA by reducing pain and ameliorating cartilage destruction [51]. On the other hand, Carlson et al. detected metabolites from steroid hormone biosynthesis in the synovial fluid of people with rheumatoid arthritis [52]. No studies have been found on the detection of alterations in ursodeoxycholic acid at the metabolomic level in animals with OA, so this metabolite should be considered in future investigations.

4-Hydroperoxy-H4-neuroprostane, also known as 14-H4-NeuroP, is a member of the class fatty acyls, superclass lipids, and lipid-like molecules and the direct parent of prostaglandins and related compounds (HMDB) [26]. Similarly, in the study by Zhao et al., it was observed that serum levels of prostaglandin estradiol2 were significantly increased in the OA group. In addition, several metabolites of the class fatty acyls and superclass lipids, such as aminobutyric acid, stearic acid, or L-carnitine, were increased [10]. Attur et al. examined plasma lipid prostaglandin E2 (PGE2) and found PGE2 elevated in symptomatic knee OA patients [53]. Similarly, Gierman et al. associated changes in PGE2 levels with the development of OA [54]. On the other hand, the study by Shi et al. and Pausinis et al. also shows changes in arachidonic acid or linoleic acid, metabolites within the same classification [2,40]. Regarding acylcarnitines of the fatty acyls class, several studies have shown changes in their concentration in the serum of animal models with OA [50,55]. Thus, changes in prostaglandin concentrations or fatty acyls could indicate the presence of OA.

There are some limitations to this study. Firstly, the sample size was small and did not allow strong validation of these potential biomarkers. Therefore, a larger sample size would be necessary in future research. Secondly, only blood samples were used, and in the future, it would be interesting to correlate serum with synovial fluid measurements to better understand the observed metabolic changes. Thirdly, the possible relationship between OA and arthritic diseases and whether these biomarkers are useful or not for identifying other forms of arthritis were not explored. Fourthly, variables other than time were not considered, and there is intra-group variability in these results; for this reason, other variables will need to be considered in the future. Lastly, it should be considered that the identified metabolites in this study should be directly evaluated in subsequent targeted studies, aiming to confirm or rule out their role in the modification of serum metabolomes in an inflammatory model of osteoarthritis.

## 4. Materials and Methods

### 4.1. Experimental Model

#### 4.1.1. Experimental Design

A prospective, experimental, randomized, and double-blinded study was designed. This study was conducted at the Hospital Universitari i Politècnic La Fe, within the animal facility of the Instituto de Investigación Sanitaria La Fe (IISLaFe), Valencia, Spain. This experimental study was approved by the Ethics and Animal Welfare Committee of the Hospital Universitari i Politècnic La Fe and authorized by the Valencian Government with 2017/VSC/PEA/00177 type 2 code, in accordance with the provisions of Article 31 of Royal Decree 51/2013.

#### 4.1.2. Experimental Trial

To carry out the study, thirty female ten-week-old Wistar rats weighing around 250 g entered the study. The rats were housed in individual cages with ad libitum access to food in an environment with a room temperature of 24–25 °C, a relative humidity of 60%, and a 14 h light–10 h dark cycle. All rats were randomly divided into three groups depending on their survival time (28, 56, and 84 days). Out of ten animals in each group, the metabolomic study was performed in six animals, and histological analysis was performed in four animals.

Osteoarthritis was induced in all 30 subjects by intra-articular infiltration of 0.4 mg of monoiodoacetate (Sodium iodoacetate^®^, Sigma Aldrich, Saint Louis, MO, USA) into the right knee. All animals were sedated with buprenorphine (0.03 mg/kg) (Buprex^®^, Indivior, Dublin, Ireland), ketamine (65 mg/kg) (Ketolar^®^, Pfizer, Madrid, Spain), and medetomidine (0.01 mg/kg) (Sedator^®^, Dechra, Bladel, The Netherlands) intraperitoneally.

Once each subgroup reached its survival times, rats were sacrificed. Thereafter, all right and left stifles were photographed for macroscopic analysis. Subsequently, the knees of four aleatory subjects from each subgroup were assigned for histological study and the other six for metabolome serum study. Blood samples were obtained from each group at day 0 (before MIA infiltration) and just before euthanasia (28, 56, and 84 days after MIA infiltration). All animals were fasted for 12 h prior to sampling. Euthanasia was performed under sedation (described above) and a subsequent CO_2_ chamber (concentration 70–100% CO_2_).

### 4.2. Obtaining the Metabolomic Results

#### 4.2.1. Sample Preparation

Serum sample preparation was performed following the protocols established in the analytical unit, as detailed below. First, 50 μL of serum, plus 150 μL of acetonitrile (ACN) and 0.1% formic acid (FA) (cold) vortexed for 30 min at −20 °C, were collected. The sample was centrifuged for 10 min at 4 °C and 13,000× *g*; the supernatant extract was collected in an Eppendorf tube and stored at −80 °C. Subsequently, 20 μL of the extract was collected and placed in a 96-well plate for liquid chromatography–quadrupole time-of-flight-6550 (LC-QTOF-6550); 100 μL of FM (Agμa 0.1% FA) + 10 μL of MIX internal standard (ISTD) (20 μM) was added. Once the plate was prepared, 5 μL of each sample was taken, and the QC was prepared. The reagent blank was prepared using the same blood collection tube used in the water study and following the same preparation procedure as the serum samples (looking for artifacts in the tube, reagents, and other materials).

#### 4.2.2. Sample Analysis

Figure 10 depicts the protocol that was followed to perform the analysis of the samples.

To avoid intrabatch variability, as well as to ensure the quality and reproducibility of the analysis, we proceeded as follows: First, a random injection order. Subsequently, an analysis of at least five quality control conditions (QC_cond_) at the beginning of the sequence was performed on the condition column and equipment (these data were not used in the multivariate analysis of the data). Finally, an analysis of a quality control pool (QC_pool_) is performed for every 5–7 samples.

#### 4.2.3. Ultra-Performance Liquid Chromatography, Time-of-Flight, Mass Spectrometry (UPLC-ToF-MS) Method

For analysis, liquid chromatography equipment was used coupled to a time-of-flight mass spectrometer. Standard procedures of the Analytical Unit lay down the chromatographic and mass spectrometric conditions, summarized below: mode positive and negative electrospray ionization (ESI); range *m*/*z*: 100–1700 Da; UPLC column: Acquity UPLC BEH C18 (100 × 2.1 mm, 1.7 μm); injection volume (V_inj_): 5 μL; column temperature: 45 °C; autosampler temperature 4 °C; flow rate: 500 μL/min; mobile phase A = H_2_O (0.1% *v*/*v* HCOOH); mobile phase B = CH3CN (0.1% *v*/*v* HCOOH).

### 4.3. Data Analysis of Metabolomic Results

#### 4.3.1. Pre-Processing of the Metabolome Data

Before performing the multivariate analysis of the data, pre-processing of the acquired data was required. This pre-processing consists of a series of processes such as filtering, molecular feature detection, peak alignment and clustering, and data normalization. Based on the results obtained in this study, several parameters were selected for each treatment. R statistical software (https://www.r-project.org/) and the XCMS library were used to carry out the data pre-processing.

At the end of the processing, we obtained two tables of “molecular features” with all the variables extracted and normalized: a table in negative mode (9712 molecular features: *m*/*z* and Rt) and a table in positive mode (4045 molecular features: *m*/*z* and Rt). Data analysis was performed from these tables.

#### 4.3.2. Analysis of the Quality of the Metabolome Results

##### Evaluation of the Response of Internal Standards

To detect possible problems in the injection or in the preparation of the samples, the variability of the internal standards, as well as intra-batch variability, was assessed for each sequence. The control chart of reserpine and leucine-enkephalin (leuEnk) used as internal standards (STDI) for the positive ESI mode is attached in Figure 11 as an example.

No tendency in the response of these patterns is observed throughout the sequence, thus assuming low intra-batch variability and correct analysis.

##### QC Evaluation

Before proceeding to the QC evaluation, the data were adjusted for the area of the internal standard LeuEnk in positive mode and reserpine in negative mode.

For the positive and negative mode sequences, the coefficients of variation (CV%) of the QCpool are calculated, and those variables with a CV ≥ 30% are removed. In this way, analytical variability is eliminated, and the remaining variables are considered to come from possible biological/metabolic variability.

### 4.4. Histological Evaluation

Following the sacrifice, the left and right stifles were dissected. A craniolateral approach of the skin and dissection of the soft tissues around the femur and tibia were performed. A 1.5 cm osteotomy proximal to the femoral trochlea and distal to the tibial plateau was conducted to retrieve the stifles. Once the stifle was isolated, periarticular soft tissues were meticulously dissected to retrieve the biological samples. After removing all the stifle soft tissue, direct visualization of the joint was performed to score the samples following the macroscopic scale described by Laverty et al. [56].

Once anatomical samples were retrieved and after macroscopical evaluation had been performed, femoral condyle samples were extracted and fixed in formaldehyde at 4%. Following fixation and decalcification with ethylenediaminetetraacetic acid (EDTA) (Osteodec^®^, LABOLAN, Navarra, Spain), paraffin inclusion and 4 µm longitudinal section cuts using a microtome were performed. Sections were obtained from three different anatomical zones: the lateral condyle, the femoral trochlea, and the medial condyle. Samples were stained using hematoxylin and eosin and Masson’s Trichrome stain. Following staining, slides were digitalized for evaluation using a specific slide viewer software (CaseViewer 2.2^®^, 3DHISTECH Ltd., Budapest, Hungary).

Lastly, microscopic evaluation was conducted using the Osteoarthritis Research Society International (OARSI) semi-quantitative scale described by Laverty et al. to evaluate matrix stain, cartilage structure, chondrocyte density, and cluster formation [56]. On top of this, researchers added an additional parameter resulting from the addition of the results of all measured variables as a total score of them altogether. The structure of subchondral bone was evaluated using semi-quantitative scales from OARSI described by Gerwin et al. [57].

#### Analysis of the Histological Results

A descriptive analysis was performed for the histologic variables. The data available were obtained from semi-quantitative scales and are presented graphically in bar charts using heat maps, which show the frequency of the values for every study period (Figure 7 and Figure 8).

### 4.5. Statistical Analysis

#### T0 vs. T28, T0 vs. T56, T0 vs. T84 Paired Analysis

For the selection of significant variables (significant differences between T0 and T28; T0 and T56; T0 and T84), a Volcano plot combining a fold change (FC) method with a *t*-test was performed. With this data analysis, the aim was to obtain an overview and to select those potentially significant variables capable of discriminating between the conditions or categories of the study.

The paired analysis was performed using a script developed in the Analytical Unit with R software. Figure 12 shows the Volcano plot obtained between T0 and T28. Figure 13 shows the Volcano plot obtained between T0 and T56. Figure 14 shows the Volcano plot obtained between T0 and T84. Note that both the FC and the *p*-value are on a logarithmic scale (log10). Variables were selected (in red) with a threshold of FC = 2 and a *t*-test *p*-value < 0.05.

## 5. Conclusions

Based on the results obtained in this study, it can be concluded that metabolomics is a promising tool to better understand the pathogenesis of OA, as MIA-induced osteoarthritis resulted in changes in the serum metabolome of rats. Furthermore, it can be observed how the metabolites that may be involved in these changes differ according to the time elapsed since the induction of osteoarthritis. Eighteen potential biomarkers were identified, predominantly from the lipid molecules, organic acids, benzenoids, and organoheterocyclic compounds classes. The lack of data on the functions of most of these metabolites helps to focus on the aim of future studies and highlights the need for further clinical studies in knee OA.

## Figures and Tables

**Figure 1 ijms-25-03158-f001:**
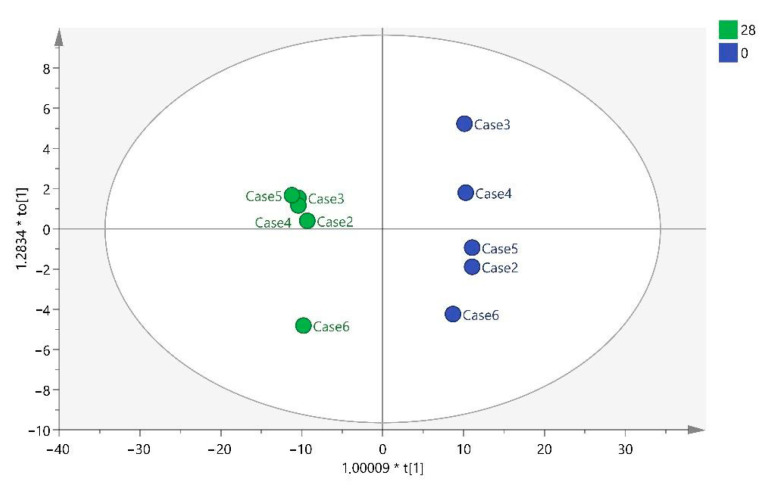
OPLS-DA T0 vs. T28. Note: multiplication (*).

**Figure 2 ijms-25-03158-f002:**
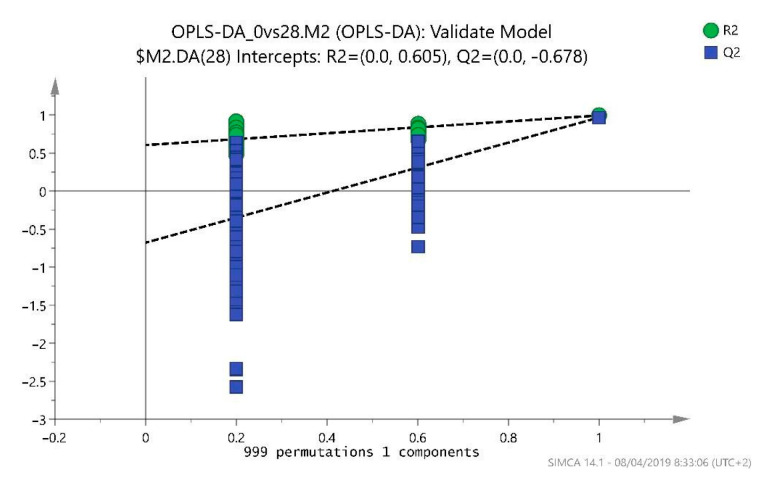
Permutation test (1000 iterations). Note: Orthogonal Projections to Latent Structures Discriminant Analysis (OPLS-DA).

**Figure 3 ijms-25-03158-f003:**
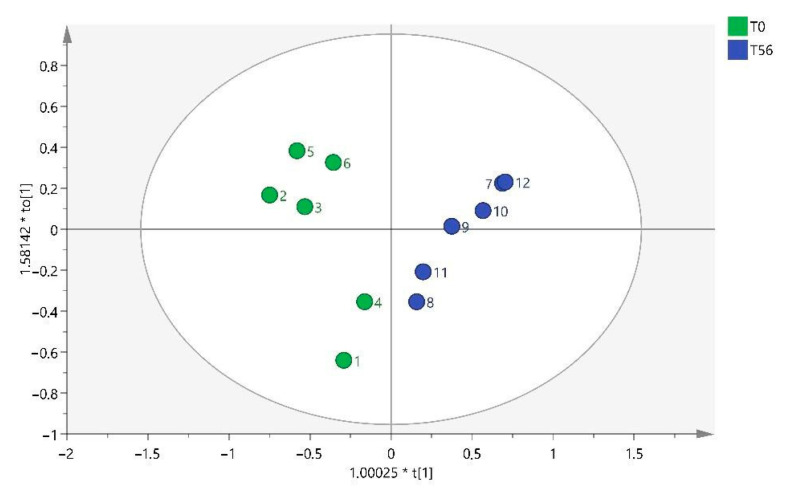
OPLS-DA T0 vs. T56. Note: multiplication (*).

**Figure 4 ijms-25-03158-f004:**
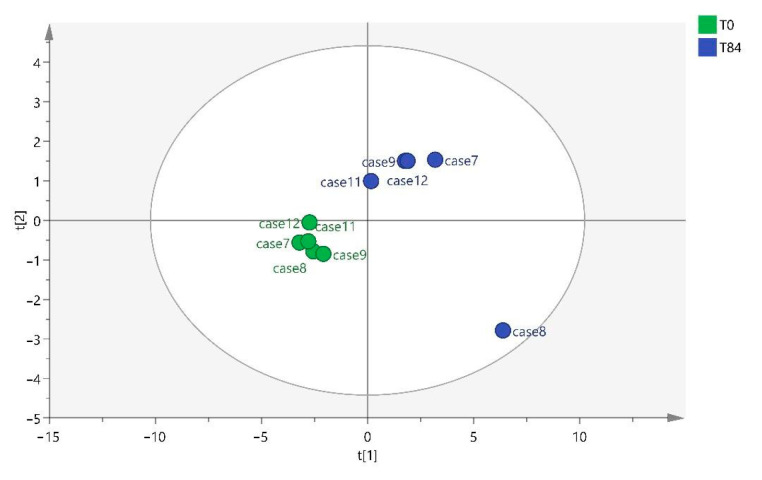
OPLS-DA T0 vs. T84.

**Figure 5 ijms-25-03158-f005:**
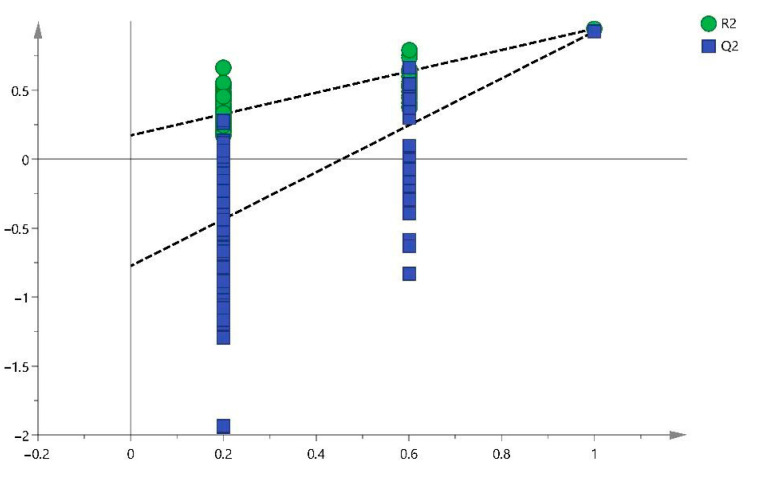
Permutation test (1000 iterations).

**Figure 6 ijms-25-03158-f006:**
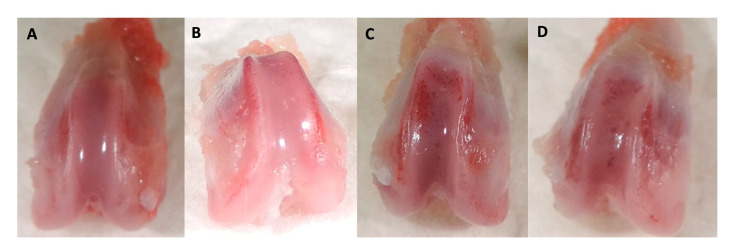
Femoral chondyle samples. (**A**) healthy sample; (**B**) 28 days; (**C**) 56 days; and (**D**) 84 days. Note the loss of articular cartilage and the presence of hemorrhages on the articular surfaces.

**Figure 7 ijms-25-03158-f007:**
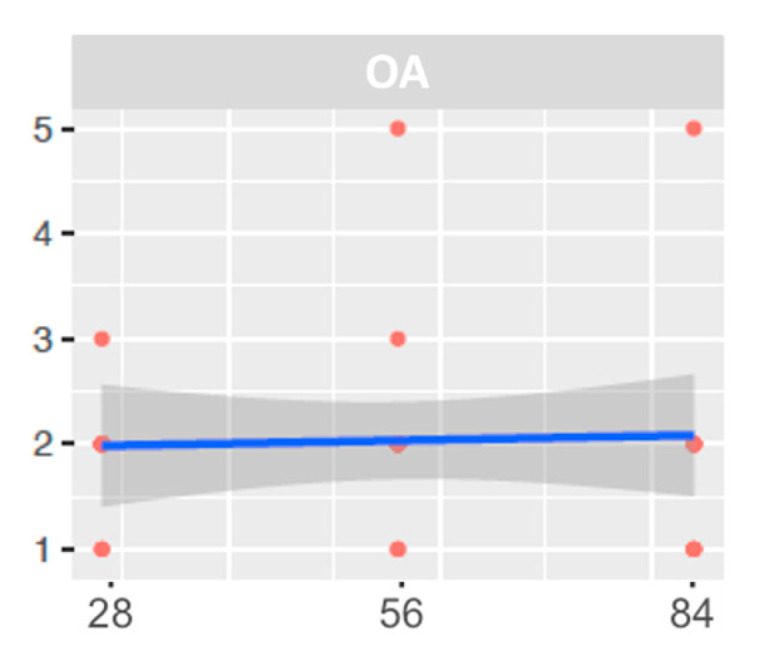
Graphical representation of the osteoarthritic stage in the macroscopic evaluation at 28-, 56-, and 84 days.

**Figure 8 ijms-25-03158-f008:**
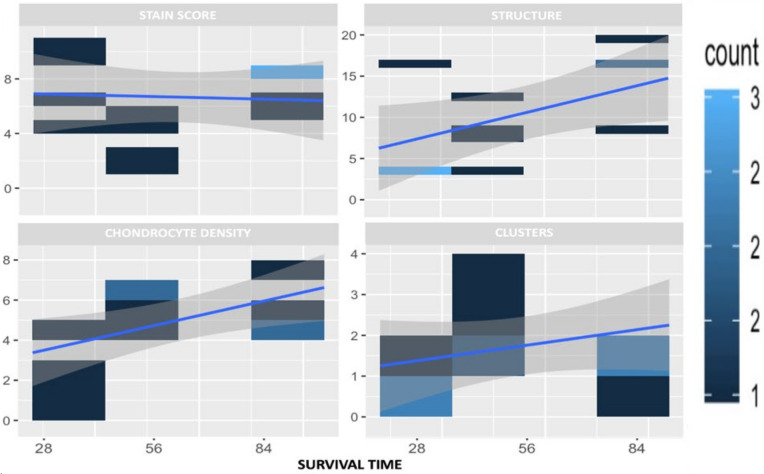
Graphical representation of the microscopic values evaluated as a function of survival time for osteoarthritic specimens, following OARSI scale values. Note: The color represents the number of times each value appears.

**Figure 9 ijms-25-03158-f009:**
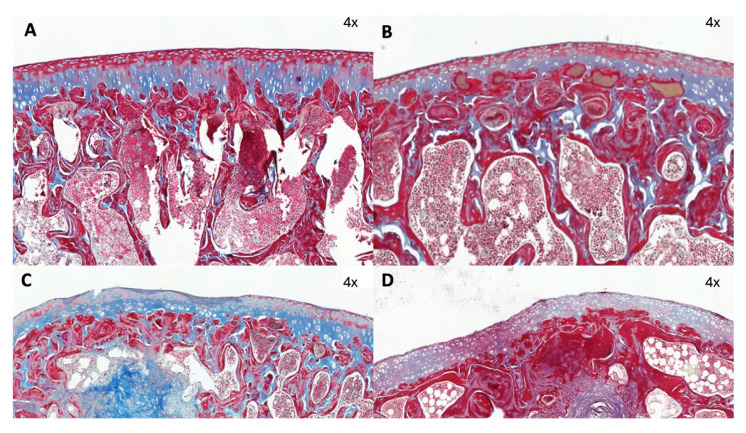
Masson Trichrome stain (4×) in (**A**) T0, (**B**) T28, (**C**) T56, and (**D**) T84 of femoral chondyles sections. The main changes observed were loss of stain intensity and irregular cell density distribution along the cartilage.

**Figure 10 ijms-25-03158-f010:**
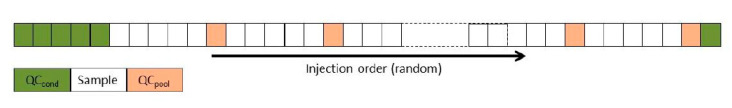
Followed scheme for the analysis of the samples. Note: quality control condition (QC_cond_), quality control pool (QC_pool_).

**Figure 11 ijms-25-03158-f011:**
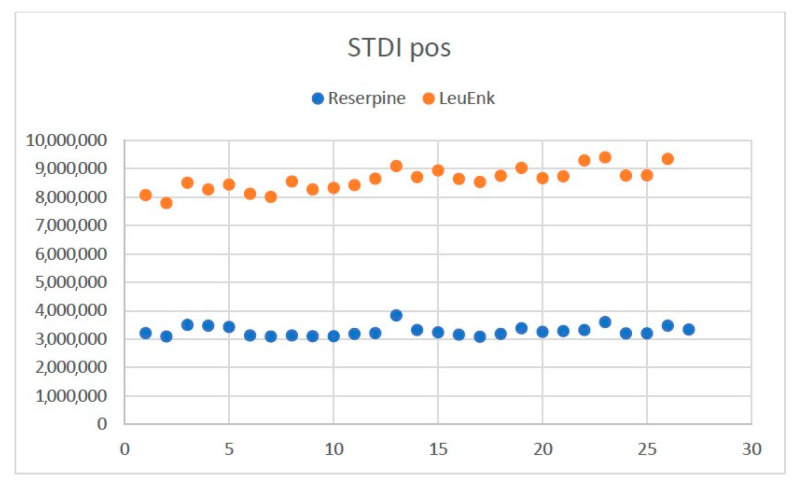
Evaluation of the reserpine response (ESI+) in the samples by injection order. Note: internal standard positive (STDI pos).

**Figure 12 ijms-25-03158-f012:**
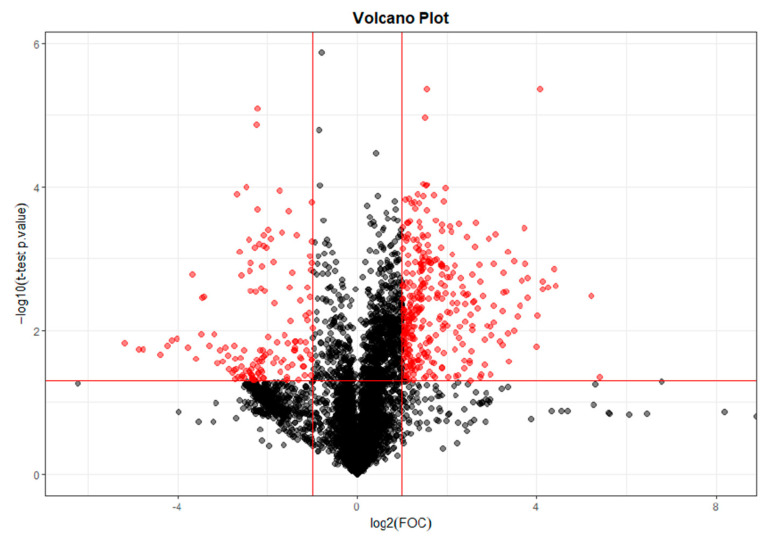
Volcano plot T0–T28. Note: Fold change (FOC). The red dots indicate points-of-interest that display both large magnitude fold-changes and high statistical significance.

**Figure 13 ijms-25-03158-f013:**
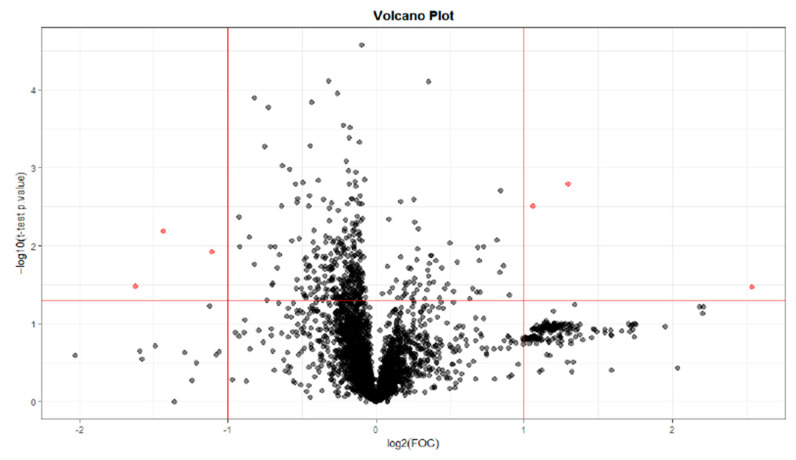
Volcano plot T0–T56. Note: Fold Change (FOC). The red dots indicate points-of-interest that display both large magnitude fold-changes and high statistical significance.

**Figure 14 ijms-25-03158-f014:**
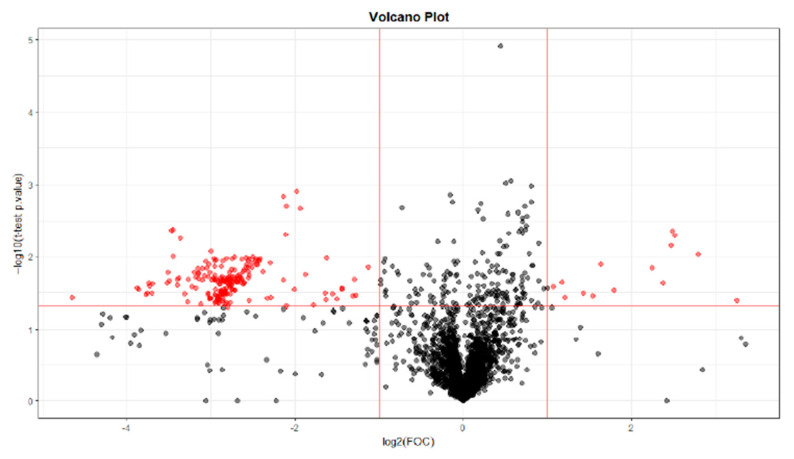
Volcano plot T0–T84. Note: Fold Change (FOC). The red dots indicate points-of-interest that display both large magnitude fold-changes and high statistical significance.

**Table 1 ijms-25-03158-t001:** Variables selected for identification at T0 vs. T28.

Variable (mz/rt)	ESI	VIP Score
377.1360487; 8.543	pos	5.547
103.0543706; 1.108	pos	4.738
395.1254806; 8.062	pos	4.337
166.0864745; 1.108	pos	4.189
120.0810237; 1.109	pos	3.800
220.1461317; 1.105	pos	2.808
171.9808059; 0.632	pos	2.491
1145.217439; 9.503	pos	2.402
352.3057184; 8.762	pos	2.314
529.3525685; 9.107	pos	2.292
148.0038081; 0.632	pos	2.127
1274.179758; 9.5	pos	2.112
335.2787707; 8.765	pos	2.046
1307.668417; 9.502	pos	1.990
1200.745803; 0.579	pos	1.957
541.3706018; 8.817	pos	1.926
1423.929102; 9.5	pos	1.921
1240.189772; 9.501	pos	1.905
1239.680395; 9.501	pos	1.890
1501.870123; 9.502	pos	1.877
1429.869007; 9.499	pos	1.860
126.0220994; 0.635	pos	1.695
1173.204232; 9.499	pos	1.584
1208.700279; 9.501	pos	1.575

Note: Positive (pos), electrospray ionization (ESI), and variable importance in projection score (VIP score).

**Table 2 ijms-25-03158-t002:** Biomarkers identified at T0 vs. T28.

Metabolites	Theory (*m*/*z*) (HMDB)	Observed (*m*/*z*)	Observed Retention Time (min)	Sub Class (HMDB)	Class (HMDB)	Superclass (HMDB)
Lactacystin	376.42	377.1360	8.543	Amino acids, peptides, and analogs	Carboxylic acid and derivatives	Organic acids and derivatives
Taurine	125.147	148.0038	0.632	Organosulfonic acids and derivatives	Organic sulfonic acids and derivatives	
		126.0220	0.635			
Styrene Oxide	120.151	103.0543	1.108	--	Benzene and substituted derivatives	Benzenoids
Tyramine	137.179	120.0810	1.109	Phenethylamines		
Setanaxib	394.86	395.1254	8.062	Phenylpyridines	Pyridines and derivatives	Organoheterocyclic compounds
Norsalsolinol	165.1891	166.0864	1.108	--	Tetrahydroisoquinolines	
Ganoderic acid V	528.7199	529.3525	9.107	Triterpenoids	Prenol lipids	Lipids and lipid-like molecules
Ganglioside GM1 (d18:0/16:0)	1519.7974	1501.8701	9.502	Glycosphingolipids	Sphingolipids	

Note: The human metabolome database (HMDB). All metabolites belong to the kingdom of organic compounds.

**Table 3 ijms-25-03158-t003:** Variables selected for identification at T0 vs. T56.

Variable (mz/rt)	ESI	VIP Score
203.2846; 6.00	neg	1.41
231.0456; 1.12	neg	0.66
279.0362; 0.80	neg	0.39
251.1023; 4.46	pos	0.12

Note: Positive (pos), negative (neg), electrospray ionization (ESI), and variable importance in projection score (VIP score).

**Table 4 ijms-25-03158-t004:** Biomarkers identified at T0 vs. T56.

Metabolites	Theory (*m*/*z*) (HMDB)	Observed (*m*/*z*)	Observed Retention Time (min)	Sub Class (HMDB)	Class (HMDB)	Superclass (HMDB)
gamma-Glutamylcysteine	250.272	231.0456	1.12	Amino acids, peptides, and analogs	Carboxylic acid and derivatives	Organic acids and derivatives
Tyrosyl-Serine	268.2658	251.1023	4.46			
Acetyl citrate	234.16	279.0362	0.80	Tetracarboxylic acids and derivatives		

Note: The human metabolome database (MHDB). All metabolites belong to the kingdom of organic compounds.

**Table 5 ijms-25-03158-t005:** Variables selected for identification at T0 vs. T84.

Variable (mz/rt)	ESI	VIP Score
595.4212; 9.168	pos	2.269
573.4081; 9.183	pos	2.266
551.3949; 9.198	pos	2.232
617.4344; 9.154	pos	2.226
529.3819; 9.213	pos	2.139
639.4473; 9.141	pos	2.118
661.4605; 9.126	pos	2.019
507.3685; 9.229	pos	2.001
683.4734; 9.113	pos	1.87
485.3556; 9.245	pos	1.822
705.4863; 9.100	pos	1.662
463.3419; 9.260	pos	1.548
820.5975; 9.294	pos	1.427
727.4989; 9.087	pos	1.34
274.2747; 7.486	pos	1.326
437.2907; 7.905	neg	1.775
391.2852; 7.905	neg	1.822
391.2125; 6.377	neg	1.715

Note: Positive (pos), negative (neg), electrospray ionization (ESI), and variable importance in projection score (VIP score).

**Table 6 ijms-25-03158-t006:** Biomarkers identified at T0 vs. T84.

Metabolites	Theory (*m*/*z*) (HMDB)	Observed (*m*/*z*)	Observed Retention Time (min)	Sub Class (HMDB)	Class (HMDB)	Superclass (HMDB)
Ginsenoside Rh1	638.8721	639.4473	9.141	Triterpenoids	Prenol lipids	Lipids and lipid-like molecules
Theasapogenol A	506.7144	507.3685	9.229			
Phosphatidic acid	674.941	705.4863	9.100	Glycerophosphates	Glycerophospholipids	
Ursodeoxycholic acid	392.572	437.2907	7.905	Bile acids, alcohols, and derivatives	Steroids and steroid derivatives	
Polyporusterone F	462.6618	463.3419	9.260			
Brassinolides	480.6771			Steroid lactones		
10-Hydroperoxy-H4-neuroprostane	392.492	391.2125	6.377	Eicosanoids	Fatty acyls	

Note: The human metabolome database (MHDB). All metabolites belong to the kingdom of organic compounds.

**Table 7 ijms-25-03158-t007:** Histological OARSI score system.

	**Microscopic OARSI Score**																						
	Stain						Structure								Chondrocyte Density					Cluster Formation			
	0	1	2	3	4	5	0	1	2	3	4	5	6	9	0	1	2	3	4	0	1	2	3
28	1	2	0	6	2	1	0	6	3	0	2	0	0	1	5	5	1	0	1	10	1	1	0
56	3	5	2	1	1	0	3	2	0	2	3	0	1	1	4	0	2	3	2	4	6	1	1
84	0	3	3	3	1	2	0	1	1	2	5	1	2	0	2	2	3	4	1	5	7	0	0

Note: OARSI score uses four different variables for cartilage evaluation. This table represents the frequency of different variable scores in three different study periods. Stain (0–6), Structure (0–10), Chondrocyte density (0–4), Cluster formation (0–3).

## Data Availability

The data are not publicly available due to confidentiality contract.

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
