# Peer review of "Changes in the Serum Metabolome in an Inflammatory Model of Osteoarthritis in Rats"

_ijms, 2024, doi:10.3390/ijms25063158_

Round 1
Reviewer 1 Report (Previous Reviewer 2)
Comments and Suggestions for Authors
Results
Regarding hystological analysis, OARSI score has not been reported in the results. Please add.
Figures
Fig 9. magnification is lacking. Please add.
Materials and methods
Please add the complete information (name, city, country, etc.) about all the materials used. Check the whole section of Materials and methods.
4.5. Statistical analysis
Only the statistical tests used for the analysis have to be reported. Please correct.
Fig 12, 13 and 14 have to be moved to the results section.
Comments on the Quality of English Language
Minor editing
Author Response
Results: Regarding hystological analysis, OARSI score has not been reported in the results. Please add. DONE
Figures: Fig 9. magnification is lacking. Please add. DONE
Materiales y métodos: Por favor, añada la información completa (nombre, ciudad, país, etc.) sobre todos los materiales utilizados. Consulta toda la sección de Materiales y métodos. HECHO
4.5. Análisis estadístico: Solo se deben informar las pruebas estadísticas utilizadas para el análisis. Por favor, corrija. Las figuras 12, 13 y 14 deben trasladarse a la sección de resultados. Al principio del artículo teníamos esas cifras en la sección de resultados, pero en revisiones anteriores otros revisores nos pidieron que moviéramos esas cifras a Material y Métodos.
Reviewer 2 Report (Previous Reviewer 1)
Comments and Suggestions for Authors
Fine with last modifications.
Author Response
Thank you very much for the previous recommendations
This manuscript is a resubmission of an earlier submission. The following is a list of the peer review reports and author responses from that submission.
Round 1
Reviewer 1 Report
Comments and Suggestions for Authors
The article under review addresses osteoarthritis (OA) as a globally impactful pathology with an incompletely understood physiopathology, often diagnosed through imaging techniques in advanced disease stages. The study's objective is to assess early changes in serum metabolome and identify key metabolites involved in an inflammatory OA animal model, I think it is a complicated objective.
The study involved thirty rats, all induced with OA through intra-articular injection of monoiodoacetate into the knee joint. Blood samples were taken from all animals and analyzed via mass spectrometry pre-OA induction and at 28, 56, and 84 days post-induction. Histological examination confirmed OA in all samples.
Results reveal identifiable changes in the serum metabolome over time, including organic acids, benzenoids, heterocyclic compounds, and lipids at T28, organic acids at T56, and lipid classes at T84. The conclusion drawn is that OA induces serological changes in the metabolome, specifically identifying 18 metabolites as potential biomarkers.
However, establishing a relationship between the identified metabolites and the sampling time was not feasible. The study suggests that these findings should be corroborated in future OA research and I hope them are done.
Author Response
REPLY: The introduction and description of methods has been improved, as proposed. In addition, English has been revised again

Reviewer 2 Report
Comments and Suggestions for Authors
English has to be revised
All the references should be checked in order to cite appropriate studies. For example:
Molecules have to be reported using subscript characters. Please check the whole manuscript.
Introduction
The introduction has to be improved in order to better explain osteoarthritis disease. Moreover, information about serum metabolites and their involvement in the pathology should be introduced.
References about epidemiology and physiopathology of osteoarthritis are not specific, except for reference 5. Please substitute with more specific ones.
References about metabolomics could correctly be cited from line 52 when authors introduce this topic.
Lines 36-38: osteoarthritis is a disease of the whole joint involving all tissues. Please better clarify this point also adding specific references.
Lines 38-43: The role of infrapatellar fat pad is completely lacking. Its role in osteoarthritis disease is emerging as an inflamed and fibrotic tissue. Please add specific references (DOI:10.3390/biomedicines10061369).
Lines 42-43: menisci degeneration is not mentioned. Please add specific references.
Lines 45-45: please specify that a previous injury represents a risk factor for developing osteoarthritis (DOI: 10.3390/jcm11154330; doi:10.1136/bjsports-2019-100959).
Lines 65-67: hypothesis should be reported after the aims of the study. Please correct.
Results
In this section only the results have to be described and not the methods.
Lines 77-78: this part could be reported before at line 75.
Lines 79-87: please move this part to the material and methods section in the statistical analysis section.
Lines 90-92: this part has to be moved to the statistical analysis paragraph from the results section.
Lines 126-134: this is a repetition. Please describe this part in the statistical analysis section of materials and methods section.
Lines 160-168: this is a repetition. Please describe this part in the statistical analysis section of materials and methods section.
Lines 103-104: please move this part to the material and methods section in the statistical analysis section.
Line 113: please delete negative (neg) from the footnote of table 1. All ESI were positive
Lines 183-184: this is a repetition. Please delete.
Line 208: 2.2. Histologic study: OARSI score has not been reported in the results.
Figures
Figure 1, 6: enlarge x and y axes text.
Figure 2: Data about T28 has to be visualized in the right part of the figure. Please correct.
Figure 10: magnification is lacking. Please add.
Discussion
This section has to be deeply improved; it looks like a list and paragraphs are not linked.
Line 256: references are missing.
Line 275: references are missing.
Line 288: authors reported “prior reports” but only one study by Pausin et al. was described. Please add the other studies authors refer to.
Line 325: specific references are lacking. Please add.
Materials and methods
Please add the complete information (name, city, country, etc.) about all the materials used. Check the whole section of Materials and methods.
Lines 360-361: Please add rats weight.
Please describe how rats where housed and in which number.
Please add laboratory conditions: temperature and light–dark-cycle.
Please add which food was provided to the rats.
Line 369: How rats were euthanised? Please add.
Line 372-374: Were fasting blood samples? Please clarify this important point.
Regarding hystological analysis, OARSI score has not been reported in the results
Section 4.4.1 Descriptive analysis should be explained.
Statistical analysis
This section is completely lacking. Please add describing all the parameters and tests used. In this way authors could remove all the introducing parts of the results subsections limiting the repetitions of the statistical procedure.
Did authors deposit our data in a public repository? Please clarify this point.
Comments on the Quality of English Language
English has to be revised
Author Response
REPLY: The introduction and description of the methods have been improved and the results are presented more clearly, as proposed. Some references cited have also been modified. In addition, the English has been further revised.
Comments and Suggestions for Authors
English has to be revised. DONE by an English language reviewer who specialises in scientific texts.
All the references should be checked in order to cite appropriate studies. For example: Molecules have to be reported using subscript characters. Please check the whole manuscript. DONE (Line 114 to 119; 142 to 143; 171 to 173)
Introduction
The introduction has to be improved in order to better explain osteoarthritis disease. Moreover, information about serum metabolites and their involvement in the pathology should be introduced. DONE. Added text from line 65 to 70
References about epidemiology and physiopathology of osteoarthritis are not specific, except for reference 5. Please substitute with more specific ones. DONE
References about metabolomics could correctly be cited from line 52 when authors introduce this topic.
Lines 36-38: osteoarthritis is a disease of the whole joint involving all tissues. Please better clarify this point also adding specific references. DONE
Lines 38-43: The role of infrapatellar fat pad is completely lacking. Its role in osteoarthritis disease is emerging as an inflamed and fibrotic tissue. Please add specific references (DOI:10.3390/biomedicines10061369). DONE
Lines 42-43: menisci degeneration is not mentioned. Please add specific references. DONE
Lines 45-45: please specify that a previous injury represents a risk factor for developing osteoarthritis (DOI: 10.3390/jcm11154330; doi:10.1136/bjsports-2019-100959). DONE
Lines 65-67: hypothesis should be reported after the aims of the study. Please correct DONE .
Results
In this section only the results have to be described and not the methods. DONE
Lines 77-78: this part could be reported before at line 75. DONE
Lines 79-87: please move this part to the material and methods section in the statistical analysis section. DONE
Lines 90-92: this part has to be moved to the statistical analysis paragraph from the results section. DONE
Lines 126-134: this is a repetition. Please describe this part in the statistical analysis section of materials and methods section. DONE
Lines 160-168: this is a repetition. Please describe this part in the statistical analysis section of materials and methods section. DONE
Lines 103-104: please move this part to the material and methods section in the statistical analysis section. DONE
Line 113: please delete negative (neg) from the footnote of table 1. All ESI were positive DONE
Lines 183-184: this is a repetition. Please delete. DONE
Line 208: 2.2. Histologic study: OARSI score has not been reported in the results. DONE (graphs added)
Figures
Figure 1, 6: enlarge x and y axes text. (NOW FIGURE 12 and 14). DONE
Figure 2: Data about T28 has to be visualized in the right part of the figure. Please correct. DONE
Figure 10: magnification is lacking. Please add. (NOW FIGURE 9) DONE
Discussion
This section has to be deeply improved; it looks like a list and paragraphs are not linked. DONE
Line 256: references are missing. There are no references, because it is a reference to the results obtained in this same study and no related literature has been found
Line 275: references are missing. DONE
Line 288: authors reported “prior reports” but only one study by Pausin et al. was described. Please add the other studies authors refer to. SENTENCE MODIFIED
Line 325: specific references are lacking. Please add. DONE
Materials and methods
Please add the complete information (name, city, country, etc.) about all the materials used. Check the whole section of Materials and methods. DONE
Lines 360-361: Please add rats weight. DONE
Please describe how rats where housed and in which number. DONE
Please add laboratory conditions: temperature and light–dark-cycle. DONE
Please add which food was provided to the rats. DONE
Line 369: How rats were euthanised? Please add. DONE
Line 372-374: Were fasting blood samples? Please clarify this important point. DONE
Regarding hystological analysis, OARSI score has not been reported in the results. DONE
Section 4.4.1 Descriptive analysis should be explained. CHANGED. These results only show as an osteoarthritis disease is present.
Statistical analysis
This section is completely lacking. Please add describing all the parameters and tests used. In this way authors could remove all the introducing parts of the results subsections limiting the repetitions of the statistical procedure. DONE
Did authors deposit our data in a public repository? Please clarify this point. The reviewers' data were not deposited in a public repository. I do not currently know the details of the reviewers. When the journal asked to choose reviewers, we chose from the database provided by the journal, experts in the subject of our research.
Comments on the Quality of English Language
English has to be revised DONE: By an English language reviewer who specialises in scientific texts.

Reviewer 3 Report
Comments and Suggestions for Authors
An interesting article about Metabolome changes in OA.
However, there are parts of the manuscript that have to be rewritten and analyzed again.
I am questioning the animal model, not that it is a suitable animal model for OA, but why the time is chosen. The chondrocytes are reduced already after 28 days. Is this an acute phase, or what would you like to show in the different time frames? Also, it is hard to interpret Figure 9, what is d. Did you only look at only one mouse? What have you done with other joints since you looked in the serum? It would be best to verify the different time frames, why these were chosen, and what is expected to be viable.
When reading the abstract, I assumed that you could follow the metabolic changes at all time points. Of course, the alteration towards the 0-time point is interesting, but I assume this would also give interesting results when the inflammation drops. Therefore, I would like to see the difference between days 28 and 84. It is hard to interpret which markers continuously change, if any; this needs to be verified in the results and discussion.
Another minor comment is regarding the sex of the rats. Are both sexes used, and does this influence the results? I know from previous publications that the normal estrogen cycle could influence metabolic parameters. The sex of the rats needs to be verified in the MM.
Comments on the Quality of English Language
I could not judge the language per se, but most of the manuscript was okay written, and I understand.
Author Response
RESPUESTA: Se ha mejorado la descripción de los métodos y los resultados, tal como se ha propuesto. Además, el inglés ha sido revisado de nuevo
Comentarios y sugerencias para los autores
Un interesante artículo sobre los cambios en el metaboloma en la artrosis.
Sin embargo, hay partes del manuscrito que tienen que ser reescritas y analizadas de nuevo. RESPUESTA: La parte de material y métodos y los resultados han sido reescritos y cambiados. Además de mejorar la introducción y la discusión
Estoy cuestionando el modelo animal, no que sea un modelo animal adecuado para la OA, sino por qué se elige el momento. Los condrocitos se reducen ya después de 28 días. ¿Se trata de una fase aguda o qué le gustaría mostrar en los diferentes marcos temporales? RESPUESTA: El momento del estudio se decidió sobre la base de nuestro trabajo previo, evaluando la degeneración articular por histología.
Además, es difícil interpretar la Figura 9, lo que es d. RESPUESTA: Hubo un error, la descripción de la figura ha sido modificada.
¿Solo miraste un solo mouse? ¿Qué has hecho con otras articulaciones desde que miraste el suero? Lo mejor sería verificar los diferentes plazos, por qué se eligieron y qué se espera que sea viable. RESPUESTA: Las rodillas de todas las ratas, no solo de una, fueron evaluadas histológicamente. Son grupos independientes de 10 individuos, los primeros 10 fueron sacrificados a los 28 días y todos fueron necropsiados. Los siguientes 10 individuos fueron sacrificados a los 56 días y los últimos 10 individuos a los 84 días. A todos los individuos se les evaluó histológicamente ambas rodillas.
Al leer el resumen, asumí que se podían seguir los cambios metabólicos en todos los puntos temporales. Por supuesto, la alteración hacia el punto de tiempo 0 es interesante, pero supongo que esto también daría resultados interesantes cuando la inflamación disminuya. Por lo tanto, me gustaría ver la diferencia entre los días 28 y 84. Es difícil interpretar qué marcadores cambian continuamente, si es que hay alguno; Esto debe verificarse en los resultados y la discusión.
Otro comentario menor es sobre el sexo de las ratas. ¿Se utilizan ambos sexos e influye esto en los resultados? Sé por publicaciones anteriores que el ciclo normal de estrógenos podría influir en los parámetros metabólicos. El sexo de las ratas debe verificarse en el MM. RESPUESTA: Todas las ratas eran hembras, esto ya está agregado en la sección de material y métodos.
Comentarios sobre la calidad de la lengua inglesa No podría juzgar el lenguaje en sí, pero la mayor parte del manuscrito estaba bien escrito, y lo entiendo.
